



# 1 The imaginary eruption. Volcanic activity through kids' eyes

Micol Todesco[1], Emanuela Ercolani[1], Flaminia Brasini[2], Delia Modonesi[2], Vera Pessina[3], Rosella Nave[4],
Romano Camassi[1]
[1] Istituto Nazionale di Geofisica e Vulcanologia, Sezione di Bologna, 40128, Italy
[2] ConUnGioco Onlus, Roma, 00081, Italy
[3] Istituto Nazionale di Geofisica e Vulcanologia, Sezione di Milano, 20133, Italy
[4] Istituto Nazionale di Geofisica e Vulcanologia, Osservatorio Vesuviano, Napoli, 80124, Italy
*Correspondence to*: Micol Todesco (micol.todesco@ingv.it)
**Abstract**
Strategies of risk mitigation become effective when citizens facing hazardous phenomena adopt rational behaviors that
contribute to lower the risk. This is more likely to occur when endangered communities share a widespread understanding of
natural phenomena and their impacts. To reach this goal, educational and outreach materials are often organized around the
descriptions of the natural process and its effects. Unfortunately, however, receiving correct information does not automatically
grant the adoption of safe behaviors. Our teaching efforts may fail because of pre-existing biases, beliefs and misconceptions.
The identification of these biases is important to plan effective educational campaigns, capable of providing the concepts that
are needed to actually inform citizens' choices about natural hazards.
In this work, we present the results of an unconventional workshop on volcanic risk that we proposed to primary and secondary
schools (ages 6-13), in Italy. The workshop is meant to explore the mental models that kids and youngsters have about volcanic
eruptions and it takes the form of a creative exercise. We asked the students to draw and write a story in four frames, describing
the onset and outcome of an imaginary eruption. All stories were then presented to the class, and always provided interesting
hints to spark discussion about volcanic processes and hazards. As a whole, the collected stories provide an interesting,
multifaceted description of volcanic eruptions and their potential impacts, as imagined by the kids. A careful analysis of this
material provided interesting insights useful to improve future outreach material and educational plans. The workshop is simple
to reproduce, even remotely, and could be easily extended to different types of hazards.
While very simple to organize, this approach grants the secure engagement of most participants and offers a very different
perspective on pupils' understanding of natural phenomena.
**1 Introduction**
The mitigation of natural risk commonly involves educational campaigns aimed at disseminating correct scientific information
among the exposed communities. A clear understanding of how natural phenomena may unfold and eventually impact our
lives is expected to favour the adoption of mitigation measures and cautious behaviour. However, the simple availability of
correct information may be insufficient. Mental models, personal experience and emotional belief play an important role in
shaping people's response to hazards. Research conducted to explore the commitment to mitigation measures against
hurricanes showed how bad habits (like leaving the windows open during a tornado) can be perpetuated by a poor
understanding of the physical phenomenon - in this case, the wrong assumption that building destruction is caused by the
pressure difference inside and outside the house (Meyer, 2009). Mental models and beliefs constitute the magnifying glass
through which laypeople will access and interpret any information regarding natural hazards (Gibson et al., 2016). Personal
experience and emotions also contribute to form risk perception and it is widely acknowledged that risk communication should
account for existing knowledge and public understanding of natural hazards, in order to target the specific needs of the
communities involved (Lacchia et al., 2020). The comparison between expert's and laypeople's mental models highlights



missing information, possible gaps and misconceptions on both sides and, most importantly, it grants a correct identification
of people's needs and expectations.
This work focuses on volcanic eruptions and their perception and targets kids and youngsters living in the Neapolitan urban
area. The town is surrounded by three active volcanoes: Vesuvius, Ischia, and Campi Flegrei. Last eruptive activity in the area
took place at Vesuvius, in 1944, when a lava plug obstructed the volcanic conduit (Sbrana et al., 2020 and refs. therein). Since
then, the volcano has been in a quiescent state, like Ischia, whose last eruption took place in 1302 (Iovine et al., 2017). Campi
Flegrei last erupted in 1538 (Di Vito et al., 1987), but this wide caldera has been giving signs of unrest since the 1950's, with
periods of remarkable seismicity and ground uplift in 1969-72 and 1982-84 (Del Gaudio et al., 2010). Then, after 20 years of
continuous subsidence, a new and slower uplift phase began in 2005 and continues to the time of writing. Ground deformation
is accompanied by minor and shallow seismicity and by changes in the composition of fumarolic gases. Observed changes led
the civil protection authorities to shift the emergency level from green (background) to yellow (scientific attention on the
phenomenon) in 2012 (Tamburello et al., 2019).
A dormancy lasting for centuries followed by decades of unrest without eruption is a common evolution for a caldera. Unlike
the case of stratovolcanoes, such as Vesuvius, even remarkable unrest phenomena may not constitute short-term precursors of
an impending eruption. However, with more than 3 millions people living in the municipality of Napoli and a volcanic risk
among the highest on the planet, this kind of volcanic pattern easily becomes a real communication nightmare. Living
memories from last Vesuvius' eruption further confuse the picture, bringing in vivid images from a very different volcanic
setting.
Given the relevance of the problem, volcanic risk perception in the area of Campi Flegrei was first tested in 2006 (Ricci et al.,
2013). Results showed that volcanic hazards were not listed among the principal concerns of a community mostly worried
about crime, pollution and corruption. Nevertheless, the people participating in the survey did consider the likelihood of
explosive eruption as moderately high. At the same time, many failed to identify the hazards posed by the caldera, which was
overshadowed by the concerns about Vesuvius. Researchers also highlighted the so-called 'optimistic bias', according to which
responding citizens tended to consider themselves less prone to severe impact than their own town (Paton et al., 2008). More
recently, a wider study was carried out to address different kinds of hazards (hydrogeological, seismic and volcanic, Avvisati
et al., 2019). Results revealed the importance of direct experience of eruption in assessing the likelihood of a future eruption
and showed that a good knowledge of the hazard does not necessarily correspond to a good knowledge of best mitigation
practices.
Within this context, we decided to focus on existing mental models of volcanic eruptions, thanks to the collaboration of the
local public schools (primary and junior high). The exploration of mental models usually takes the form of interviews
(Skarlatidou et al., 2012) or face to face surveys accompanied by follow up questions (Lacchia et al., 2020). However,
considering our particular target, we opted for a different approach. To engage participants, we proposed a creative writing
and drawing exercise, asking them to describe a short story featuring an eruption and its consequences. We collected
approximately 200 stories that depict a range of rather plausible scenarios for this volcanic area. We analyzed all stories in
detail, identifying the spatial and temporal frames in which kids place their eruption, as well as the accompanying words and
feelings.
While certainly not comparable to the results from more structured approaches, our exercise provided valuable insights on
widespread expectations and useful hints for future outreach materials.



## 2 The project

"The imaginary eruption" is an activity promoted within the EDURISK framework (www.edurisk.it), a long-term educational project with the aim of promoting educational itineraries for risk reduction for schools of all grades, with particular attention to ages 6 to 13 (Pessina and Camassi, 2012).

The activity involved ten school districts (8 in the Neapolitan area and 2 from non-volcanic regions) and was carried out during two school years, in 2018 and 2019. A total of 25 classes participated in the activity, 13 from primary schools (6-10 years old) and 12 from secondary schools (11-13 years old), for approximately 500 kids. In 2020, we proposed the same workshop to the schools on the island of Stromboli. Due to the restrictions related to the COVID-19 pandemic, we held this activity remotely, via videoconference. On the island, we gathered 11 stories on the island, 9 of which completed with drawings, and 2 featuring only written text. Among the complete stories, 6 were from primary school and 3 were from secondary school. The analysis presented below focuses on the earlier workshops run in presence, while results obtained in Stromboli are discussed aside. A list of the schools involved is provided in Table 1.

**Table 1: List of attending schools.**

| School district and name | Classes | School level |
|---|---|---|
| IC 3 De Curtis Ungaretti, Ercolano (NA) | 3A, 3B | Primary |
| IC 2 F. Giampaglia, Ercolano (NA) | 4 | Primary |
| IC 6 Quasimodo Dicearchia, Pozzuoli (NA) | 4D | Primary |
| DD Scafati 1, Scafati (NA) | 2A, 2B, 2C, 3B, 4A, 5D | Primary |
| IC Bonati, Bondeno (FE) | 3A | Primary |
| IC San Rocco di Marano, Napoli (NA) | 3A, 3B<br>1A | Primary<br>Secondary |
| IC 5 Testoni Fioravanti e Federzoni, (BO) | 3A, 3B<br>3E | Primary<br>Secondary |
| IC 3 Rodari-Annecchino, Pozzuoli (NA) | 1C, 1F | Secondary |
| IC 3 CD S. Gaetano-Gadda, Quarto (NA) | 1A, 1D, 1F, 2C, 2E, 2G | Secondary |
| IC S De Nicola Sasso, Torre del Greco (NA) | 1B, 1D | Secondary |
| IC 2 De Amicis-Diaz, Monteruscello (NA) | 1C, 1E | Secondary |
| IC Isole Eolie - Stromboli | Multi-age class<br>Multi-age class | Primary<br>Secondary |

The activity featured an initial phase of direct interaction with the attending students. During this workshop, we guided the stories' realization, as better specified below. We introduced "The imaginary eruption" as a creative exercise rather than a science essay. This was important to collect stories that probed the kids' mental model rather than reflecting lessons' contents. We stressed the absence of a formal evaluation of the "correctness" of the description and suggested the possibility of fantastic settings or characters. At the end of the workshop, participants shared and discussed their stories. Each tale provided many opportunities to discuss eruptions, volcanic phenomena, various hazards, and their mitigation.

After the workshop with the students, we held three meetings with the teachers. During these encounters, we adopted participatory techniques and explored the thoughts and feelings that emerged from the collected stories. In a few cases, the teacher also participated in the same laboratory as the kids, producing their own stories about the eruption. The meeting with the teachers explored different aspects of volcanic risk and resilience: during the first meeting, we addressed the environment and its relations to volcanic hazards and risks. The teachers explored the visible and invisible features characterizing the landscapes where they live and work. The analysis provides clues on volcanic risks and what amplifies or mitigates them. The second meeting focused on responsibility, community, resources, and problems: the group assessed how to prepare and what





to do to mitigate the risk. Finally, the last meeting revolved around resilience, identifying the times and means to share
information and understanding.
We finally used the considerations raised during these discussions to plan future outreach activities related to volcanic hazards.

**3 The workshop**

Students worked in pairs or small groups, and their assignment was to invent a story in four frames, each featuring both
drawings and a written description. The materials required for the story's realization included: 4 sheets of paper (A4); pens,
pencils, and colors; eraser and sharpener; scissors. We asked participants to cut the sheets into a square (21x21 cm) destined
for the drawing and use the remaining rectangular stripe for the written text. Once the material is ready, we provide indications
to start, and we also specify the time (approximately 15 minutes) allowed for the realization of each frame. We instructed
participants to realize one frame at a time, following simple indications often offered in terms of guiding questions. An
important detail is that students were unaware of participating in a volcanic risk project and that the stories should have a
volcanic eruption as their main theme.  Instructions were as simple as possible to allow ample creative freedom but were
needed to focus on volcanic eruptions and make the drawings comparable.
The first frame sets the story's scene. Students had to describe the main characters involved, the general setting, and the
environment in which they move. We only gave the constraint that a volcano should be present. Guiding questions for this
frame could include: *Our character(s) live(s) near a volcano: what kind of place is it? Who is the protagonist? What is she/he*
*doing? How does she/he feel?* Only when the first frame was finished (or when the allotted time passed), we provided
information on the successive step.
In the second frame, something unusual happens with the volcano. Participants had to describe what was going on and the
characters' reactions. The guiding questions inquired if someone noticed the changes or took action: *the volcano is doing*
*something unusual; perhaps it's waking up, What does the volcano do? Does the character see that? Does he/she talk about*
*that with someone? Do they do something about it? How do they feel about it?*
The third frame focuses on the eruption. We asked participants to describe the volcanic event and tell about its impact on the
surroundings and how it affected the characters. Guiding questions included: *The eruption begins, what does the protagonist*
*do? What do other people do? What is happening around them? How do they feel?*
The fourth frame is the story's epilogue: the eruption is over. Participants described the new setting, where the protagonists
are now, how much time had passed since the eruption. Possible guiding questions were: *The eruption is over: where are the*
*characters now? What do they do? How do they feel?*
An example of a full story is provided in the Supplement.

**4 Results**

We collected 190 stories (2 of which without written text), with text and drawings (26 black and white, 148 in color). The
stories often describe realistic settings, but many contain imaginary situations or magic characters and events. Stories take
place in all sorts of locations, from the close neighbourhood to far, exotic places, as distant as other planets. Among the
protagonists we find kids, youngsters, adults, superheroes, animals. While stories tend to have a happy ending (in 163 cases),
most of them acknowledge severe destruction caused by the volcano. Some stories (25) have a dramatic conclusion, and a few
culminate with the protagonist's death.

**4.1 The stories' language**

We scanned all the stories and digitized the written descriptions (available for 188 stories) to perform a simple text analysis.




Text mining was performed with the R software (R Core Team, 2020), using a specific package (tm), and involved a
preliminary manipulation to remove the punctuation, extra white spaces, and the common words (or stopwords), like articles
or prepositions, which are not expected to bear specific information for the analysis. A collection of stopwords is available for
the Italian language in the R function we used (tm_map), but we added a few more that emerged from a first text review (as
reported in the caption of Figure 1). The resulting corpus contained 3428 terms that were used with different frequencies. More
than 1950 words were used only once. As expected, the most common term is vulcano [volcano], which is mentioned 581
times. Focusing on the other terms, the three most frequent words are casa [home], with 147 occurrences, Vesuvio, mentioned
127 times, and lava, which appears 124 times. Figure 1 shows an histogram of the 20 most used words (translation in the
caption). A more general idea is provided by the word cloud.

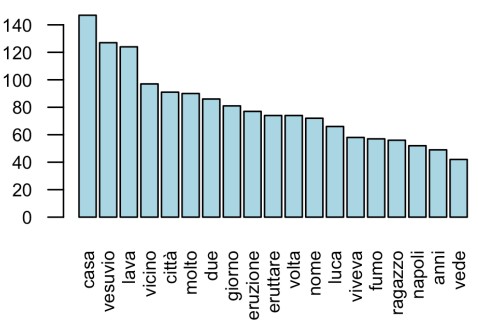
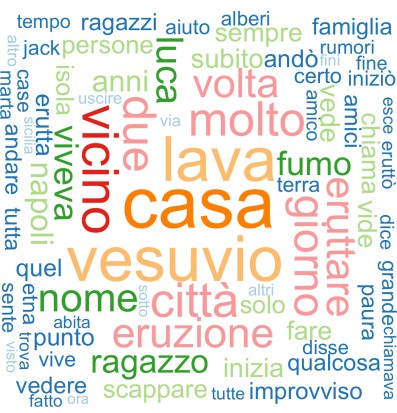

**Figure 1: Frequencies of the most used terms in the written descriptions of the imaginary eruption and associated word cloud.** The
meaning of the terms are as follows: casa [home], Vesuvio [Mt. Vesuvius], lava [lava], vicino [nearby], città [city], molto [much], due [two],
giorno [day], eruzione [eruption], eruttare [to erupt], volta [time, turns], Luca, viveva [lived], fumo [smoke], ragazzo [boy], Napoli, anni
[years], vede [she/he sees], subito [now], inizia [begins], fare [to do], vide [she/he saw]. Stopwords added to the original list and not included
in the count are: perché [why], così [therefore], poi [after], allora [then], cosa [what, thing], quindi [therefore], però [however], po' [a bit],
dopo [after], mentre [meanwhile], lì [there], quando [when].

The digitized text allows us to verify how many times specific words are used. We can see how many times death or salvations
are explicitly mentioned in the descriptions searching for the recurrences of the words related to death (including the
declination of the verb to die) and those related to survival or salvation. Frequencies of each term are listed in Table 2. The
simple frequency of these terms does not reflect the actual meaning of the story, as it does not account for possible negation
("I did not die" or "they did not survive"). Table 2 shows that terms referring to salvation are slightly more mentioned than
those referring to death.
The same exercise can show how many times the words girl(s) (ragazza) and boy(s) (ragazzo) are mentioned. The search
included words for baby girls (bambina) and baby boys (bambino). The masculine term ragazzo (56) is almost twice more
frequent than the feminine ragazza (33). Baby boys and girls are less mentioned, but the difference between them is much
smaller (bambino (16), bambina (14)). The feminine plural terms are not common (ragazze (7), bambine (0)), while the plural
masculine, which in Italian may refer to both genders, is more frequent (ragazzi (34), bambini (15)). These kids and youngsters
are often protagonists of the stories and are commonly surrounded by their friends and family or by other people. Figure 2
illustrates the frequencies of the terms related to the people who are protagonists of the stories (specific Italian terms are listed
in the figure caption).




**Table 2: Recurrences of terms related to death and to the verb to die (*morte, moririe*) and survival (including references to the words safe, alive, and the verbs to save, to survive) in the written descriptions of all collected stories.**

| Word | Frequency | Word | Frequency |
|------|-----------|------|-----------|
| morte | 7 | salvo | 11 |
| morì | 7 | salvi | 8 |
| morti | 7 | salvò | 8 |
| morto | 5 | salvati | 7 |
| muore | 4 | salvato | 6 |
| muoiono | 4 | salvarono | 5 |
| morirono | 4 | salva | 4 |
| morta | 3 | salvano | 4 |
|  |  | vivi | 2 |
|  |  | sopravvissuti | 1 |
|  |  | salvata | 1 |
| **Total** | **41** |  | **57** |


The written descriptions shed light on the words used to describe the volcanic phenomena and products. The imaginary volcanoes mostly emit lava (124) and smoke (57) (Figure G), but products of magma fragmentation are also described as ash (19), stones (18), and lapilli (11). Magma, volcanic gases and dust are also mentioned a few times.


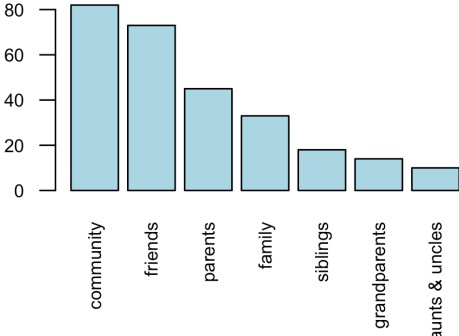


**Figure 2: Frequency of words related to friends, families and communities.** The categories shown in the figure include the following Italian terms: 'community': cittadini [citizens], abitanti [residents], persone, gente [people]; 'friends': amico [friend] and compagno [companion]; 'parents': mamma [mum], madre [mother], papà [dad], padre [father]; 'family': famiglia [family]; 'siblings': fratello [brother], sorella [sister]; 'grandparents': nonna [grandma], nonno [grandpa] (note: the Italian language does not have a formal expression for grandmother or grandfather); 'aunts & uncles': zia [aunt], zio [uncle]. Both singular and plural, and masculine and feminine are always counted.




Other words that may be of interest in this analysis are those referring to professional figures that may be related to the
assessment and the management of volcanic crises. Terms associated with these professional roles include scientists (16, one
of which female), volcanologists (17, one of which female), geologists (8, four of which female), firefighters (12), civil
protection, police, rescuers, and the mayor (2). The corresponding Italian terms are: ‘volcanologist’: *vulcanologo*; ‘scientist’:
*scienziato*; ‘geologist’: *geologo*; ‘firefighters’: *pompieri, vigili del fuoco*; ‘civil protection’: *protezione civile*; ‘police’: *polizia*;
‘rescuers’: *soccorsi*; ‘mayor’: *sindaco*. Both singular and plural, and masculine and feminine are always counted.
Words can also tell us something about how the characters feel about the events. Figure 4 shows the most common words
related to sentiments: fear (79) includes terms such as *paura* [fear] (31), *panico* [panic] (10), *impaurito/a* [scared] and various
forms of the verb *spaventare* [to be scared]; *happiness* (22) includes words like *felice* [happy] (26), *contento* [glad] (20), or
*felicità* [happiness] (1).

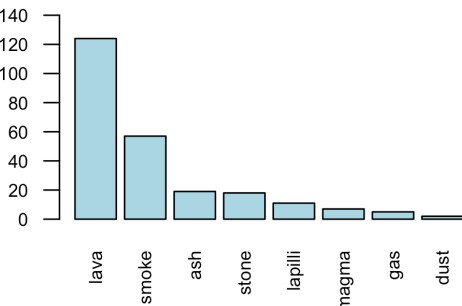


**Figure 3: Frequencies of words used in the description of volcanic products.** The corresponding Italian terms are: ‘lava’: lava; ‘smoke’:
fumo; ‘ash’: cenere; ‘stone’: pietra; ‘lapilli’: lapilli; ‘magma’: magma; ‘gas’, gas; ‘dust’: polvere. Both singular and plural are always
counted.

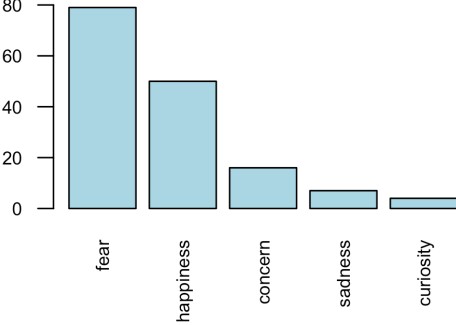


**Figure 4: Frequencies of terms related to sentiments like fear, happiness, concern, sadness and curiosity.** The corresponding Italian
terms are as follows: ‘fear’: paura [fear], panico [panic], impaurito [frightened], spaventato [scared], spaventare [to scare]; ‘happiness’: felice
[happy], contento [glad], gioia [joy]; ‘concern’: preoccupato [worried], preoccupazione [concern]; ‘sadness’: tristezza; ‘curiosity’: curioso
[curious], incuriosito [intrigued]. Both singular and plural, and masculine and feminine are always counted. In the case of verbs, different
tenses and persons are considered.






**4.2 The frame contents**
**4.2.1 Frame 1 - The protagonists and the environment**
In 148 stories (78% of the total), the main characters are real people, often representing the authors themselves or other kids
slightly older than them. In 43% of the stories the protagonists are referred to as boys (55 times) or girls (27 times), but in
many stories one or more adults are present. Adults are mostly men (77% of adults) and are often identified through their
employment (e.g., farmer, scientist, explorer, rock star, hunter, lumberjack, astronaut, soccer player...). Scientists are
mentioned as leading characters in 12 stories, 4 times as volcanologists, and 2 times as geologists. In 2 stories, the protagonists
are related to civil protection. In a smaller number of cases, the adults are identified through their family relations with other
characters (grandpa, husband, wife, mother, father). Often the protagonists are accompanied by friends or pets and in a few
stories the animals are the leading characters. Fantasy characters appear in 54 stories and include princes and princesses,
magicians and fairies, cartoon characters and superheroes, aliens, gods, and pirates. In a few cases, the volcano itself becomes
a character, with anthropomorphic features.

The volcano is always represented as a conic mountain, generally rather small. Sometimes it is depicted with two peaks,
mimicking the profile of Mt. Vesuvius surrounded by Mt. Somma, the remnant of an ancient caldera structure (interestingly,
Mt Somma is often represented as a second volcano, with its own crater). Only one story features a submarine volcano. The
volcano can be a real one, with Vesuvius being the most common choice (named in 32 stories), followed by Etna (13) and
Solfatara (2). Stromboli, Vulcano, Ischia and Monte Nuovo are also mentioned once. In a few cases, the volcano has a fantasy
name, while often it is nameless. Explicit reference to the city of Naples is also present in 17 stories. Other localities mentioned
include Sicily (8), Pompei (4), Ercolano (3), Torre del Greco (2). Exotic settings are also frequent, with reference to Hawaii
(4 stories), Arequipa (Perù), Australia, Hollywood, Los Angeles, Paris, Alaska, Russia, Texas, Norway, the Caribbean. The
volcano can be on an island (18 stories) or surrounded by woods (8), in the countryside (9). Exotic environments include the
savanna, the Indian jungle, or the desert. Four stories are set on other planets.
Inhabited areas are rare and often represented by a single house (17 stories) usually built near the volcano. In a few cases, there
is specific mention of a small town or a village nearby (13 times) and only 7 stories mention a city.
A few examples of the first frame are collected in the Supplement (Frame 1).

**4.2.2 Frame 2 - Something happens**
The most frequent sign of volcanic unrest is ground shaking (in 48 stories), sometimes described in terms of seismicity (the
word earthquake is used 12 times, with a couple of references to its magnitude). Other stories use generic terms like tremor
(20 times), or ground movements (15 times). Opening of cracks, landslides or building collapse are also mentioned  (Figure
5).  Another common signal of volcanic unrest is the presence of smoke on top of the crater, which is mentioned 44 times in
this frame. Based on both the written description and the drawings, the word smoke is used to intend volcanic gases (which
are explicitly mentioned only 2 times). Another reference to degassing activity is the smell (sometimes specifically sulphur
smell), that is mentioned in 8 stories. Volcanoes also make noises, which are mentioned in 33 stories. Other signs of unusual
behavior refer to actual eruptive processes, like the emission of lava (20), or various ejecta (stones and rocks, lapilli, and
volcanic ash, mentioned in 14 stories).





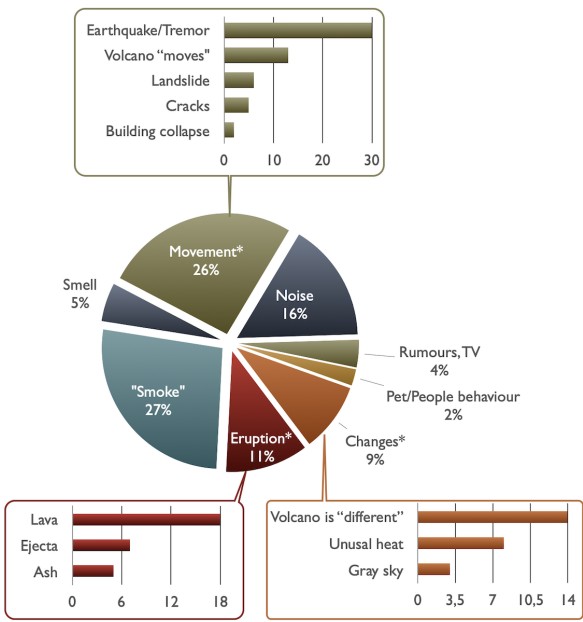

**Figure 5: Signals of unrest mentioned in the collected stories (frame 2).**

Eruptions or explosions are mentioned 47 times at this stage of the story. In a few cases, the signals that something is going on are changes in the volcano's color or appearance, or the characters perceive an anomalous heat. Flames and burned vegetation are mentioned in a few cases, while animals detect the unrest in a couple of stories. In most cases, evidence of volcanic unrest is obvious enough for the protagonists to notice themselves. The only (indirect) reference to sensors installed to monitor the volcano is a seismogram drawn in one of the stories, while other 3 mention the magnitude of the earthquake, which presupposes the presence of seismometers in the area. In all other stories (the great majority) the signals from the volcano are easily detected by residents, with no need for monitoring instruments. Sometimes the characters learn (or have confirmation) that something is going on by watching the television (8 stories).

Most characters are frightened by the unrest (39 stories mention fear in this frame). Common reactions include talking to other people (50% of the stories) to warn them, but also to ask for explanation or seeking help. In about 30% of the stories, the protagonists have a companion, and sometimes they may talk to each other about what is going on. Talking to other people involves friends (26 stories), family (22), scientists (12), the community (either everybody, or the neighbors) (10). The authorities (police, firefighters, civil protection, but also the mayor, the professor, the director…) are called upon in 9 stories, while 5 times the protagonists refer to a wise, old character for advice.

In a few stories the protagonists try to warn others but are not believed or receive no answer. In one story, the volcano itself talks to the protagonist warning him to go away. In another story the warning comes from the mailman. In most cases, the protagonists realize that an eruption is possible, sometimes thanks to the opinion of others. Only in a few cases, the characters fail to recognize the danger or consider the signals a normal feature of the volcano that raises no worries.

In the majority of the stories, this frame does not specify whether the characters are going to take action. Only 22 stories (12% of the total) explicitly refer to leaving the place because of the impending danger, while in 15 cases, the character goes or remains home, or seeks shelter, waiting for further development. While most of the protagonists are worried, some are fascinated by the unusual phenomena, or look for answers and willingly move toward the volcano (17 stories). In a few stories (12), the characters seek a remediation for the problem: sometimes it's a magical intervention, while others refer to some kind



of authority to 'fix the problem' (usually, in an unspecified way). A small number of protagonists take actions to mitigate the
hazard, obstructing the volcanic vent with rocks (or even a cork), or pouring water inside the crater.
Examples of frame 2 are provided in the Supplement (Frame 2).

### 279    4.2.3 Frame 3 - The eruption

The eruption is usually sudden: the terms *improvviso* [sudden] and *improvvisamente* [suddenly] are used 26 times in this frame,
and *subito* [immediately] appears 8 times. The event may be described as an explosion or a blast (32 times). Based on drawings,
it is usually a small-scale event, whose impact is generally confined to the upper portion of the volcanic cone. It commonly
involves the emission of lava (mentioned in the text 60 times), but the Strombolian ejecta are present in several drawings.
Written text mentions ash, rocks and lapilli (23 in total, for this frame). Flames are also mentioned about 10 times, as a result
of fires set by the incandescent eruptive material. Only a few stories actually depict a major explosive eruption that occupies
most of the drawing area and generate clouds of ash that could represent pyroclastic flows. The terms pyroclastic flow, or nuée
ardente, are never used in the written descriptions.
As the volcano erupts, people scream and run away, but also watch, in a few cases with fascination, while many are frozen in
fear. Many characters just watch, gathering in the streets but without leaving. In several stories, the onset of the eruption is the
time when the characters begin to worry and start wondering what to do next. In a few cases, this is the frame where people
are warned about the danger, often by word of mouth, and only in very few cases through official actions (a siren, or police or
civil protection authorities alerting the population). The word evacuation is only mentioned twice.

The most common reaction to the eruption is to take the flight (83 times, in this frame, 60% of the stories considering both
text and drawings), mostly by running (Figure 6).

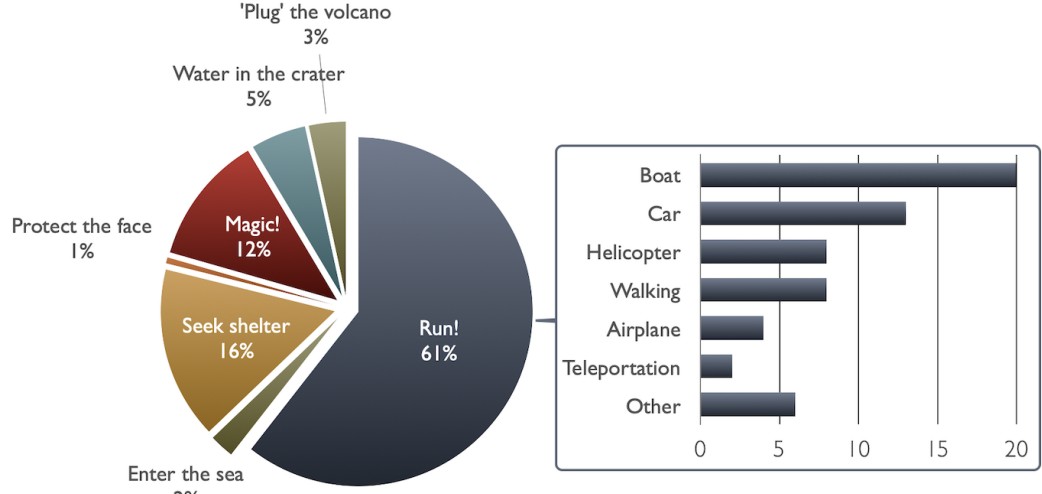


**Figure 6: Characters' actions and response to the eruption (frame 3).**

Many escape on a boat or a raft (20), others rely on their car. Airborne vehicles are also popular, helicopters in particular but
also airplanes, air balloons, or fantastic vehicles. Only a few leave the town by train or bus, or use animals. Many seek shelter,
often returning to their home (26 times), or entering into the sea. Also in this frame, a few bravely take some sort of action to
mitigate the hazard (34 times), either through magic intervention but also by pouring water into the crater, or by throwing
rocks into it, to obstruct the conduit. Rarely the action is taken by the community, or by public authorities.



Others call for help (21 times), sometimes hopelessly.
Text and drawings also reflect the damage (mentioned 34 times) to the environment (burned vegetation) and to infrastructures
(houses, mostly, but also cracks or lava interrupting roads). In 9% of the stories the destruction is pervasive and impacts the
entire city. In 11 stories someone dies in this frame, in a few cases the protagonists themselves do not survive. Examples of
the third frame are collected in the Supplement Frame 3.

### 4.2.4 Frame 4 - The epilogue

This frame shows how things ended, and the participants imagined a rather wide spectrum of possible outcomes: in some
stories the old life resumes, as if nothing happened, thanks to magic interventions or because it turns out it was just a dream.
In other cases, nothing will be the same ever again, because someone died or because it was necessary to move and live
elsewhere. Most of the stories end well (58%): the characters survive, perhaps a little buttered (bruises or wounds are
mentioned; in one story the two characters end up with a headache). They often contribute to save their community, and
happily celebrate the end of the eruption. The relief is usually burdened by the damage caused by the volcano: even though
the characters are alive, in several stories (28%) they face destruction and losses. Destruction is mentioned 38 times in this
frame, and it affects both the environment (trees, animals) and urban infrastructures (buildings, roads), that are burned (15
stories) or covered in ashes (11 stories).
The disconsolate assessment of the devastation may be accompanied by the idea of reconstruction (13% of the stories), which
may take place either right away or after a long time. The protagonists may or may not be directly involved. Some choose to
rebuild elsewhere. Sometimes, the reconstruction is carried out while the characters are away, in a safe place. Other times
(11%), the reconstruction is described as a community effort acted by everybody, or by the city, or by the inhabitants. Only in
a few cases (3%) specific categories are mentioned to be in charge of reconstruction (masons, firefighters). Sometimes the
community that builds a new life after the disaster is limited to the characters and friends or relatives. Rescue teams of some
sort are mentioned only 6 times in this frame.
The need to move away is again mentioned (24% of stories) and only some of the protagonists (7%) envisage returning home,
perhaps after a long time. A small number of stories (10%) do not end well, and remind us that things can go really bad. Death
is explicitly mentioned 38 times in this frame, and in 15 stories the protagonists themselves die, sometimes in the heroic effort
to save their community, other times in loneliness or because nobody survives. In a couple of cases, the death is not directly
related to the eruption, but due to indirect or independent causes (heart attack, being hit by a firefighters truck, and even by an
atomic bomb, totally unrelated to the story).
A few stories do not provide details on the epilogue and simply state that the eruption ended, without further comments. The
story simply ends because the volcano "turns off" (or it rains, and this stops the eruption). There may be an acknowledgment
that everything is "burned", but without information on what happens to the characters. In one story, the houses are only
covered by ashes, but still habitable. In another one, the bad ending features widespread destruction and scientists who do not
know what to do.
Examples of the fourth frame are collected in the Supplement (Frame 4).

### 4.3 Stromboli

The same workshop was later proposed to the student living on the island of Stromboli (Aeolian Island, north of the coast of
Sicily). This offered us the opportunity to collect stories from a context that is very different in terms of its geographic,
volcanic, and social settings. Stromboli is an open-conduit volcano, usually characterized by a mild Strombolian activity
(which takes its name from the island itself) and features continuous degassing and repeated explosions ejecting materials up
to tens or hundreds meters above the crater. Lava flow may occasionally form along the deserted slope of the volcano. This
persistent activity typically impacts only the summit of the volcano and is considered one of the main touristic attractions on



the island. This behaviour is sporadically interrupted by greater explosive events, known as paroxysms. The Imaginary
Eruption workshop was carried out in 2020, right after two paroxysmal eruptions took place in 2019, on July 3 and August 28
(Giordano and De Astis, 2021). Both events could raise an eruptive column of several kilometers and generate pyroclastic
flows that rushed down the deserted slope of the island to reach the sea. The two paroxysmal eruptions occurred without
noticeable precursors, were unusually close in time and the first one caused one causality, shaking the busy touristic season
that revolves around guided tours of the volcano's summit. Residents had to face the fear of both volcanic eruption and
economic disruption at the same time. In this context, we planned a workshop for the spring 2020 but, because of the COVID-
19 pandemic, we couldn't travel to the island. We did not want to miss the opportunity to get in touch with the students and to
offer them a safe space to discuss volcanic eruptions and their consequences. We therefore adapted the workshop to make it
suitable for remote fruition. Participants connected from their home and since it was not possible to organize the work in pairs,
each student elaborated her/his own story. We allowed longer times to work and to discuss the stories at the end. The workshop
was therefore organized in three days, with one session of 1 hour each day. The attending students were 13, from both primary
and secondary schools but we could collect only 9 stories featuring both drawing and written descriptions. This is a very small
number to allow for wide considerations. Nevertheless, we do consider these stories of interest for the peculiar circumstances
they reflect and we therefore provide here a terse description. Most stories are set on the island itself which, in two cases, is
the actual protagonist. The impending eruption can be announced by rock fallout (which in one case set the vegetation on fire),
small tremors, or even small eruptions. In one case, the eruption is announced by an "air pocket". The eruption is often a
typical strombolian eruption, with a lively launch of ash and scoriae, sometimes associated with a lava flow. In one case the
eruption begins under water, while in another one pyroclastic flows and their destructive power are mentioned. The stories
mostly feature a happy ending, with the volcano returning to its usual behaviour and inhabitants can resume their usual lives.
In a few cases, however, consequences are more serious and involve injured people or imply leaving the island and friends.

## 5 Discussion

The Imaginary Eruption provides a composite portrait of volcanic eruptions and their impact, as perceived by the kids and
youngsters, mostly from the urban area of Napoli. As a whole, it is a rather accurate portrait featuring many realistic features
that can be expected during a volcanic event. Collected stories provide a wide range of plausible eruptive scenarios. A
comparison with those envisaged by the scientific community reveals a few gaps and discrepancies that could inform future
outreach programs.

### 5.1 The volcano and its activity

In all collected stories, the volcano is an obvious geological feature of the landscape and the eruption invariably takes place at
the summit of the cone. A caldera setting is never mentioned, nor the possibility of new vents opening along the slope of the
volcano, or elsewhere. Pupils and students are commonly very passionate about volcanoes, and this passion is generally
accompanied by a good knowledge of different volcanic structures and phenomena. While many of the kids involved are
certainly aware about calderas and their behaviour, the choice of representing classic volcanic cones reflects the conventional
image that we all picture when we think about volcanoes. Some drawings explicitly refer to Vesuvius, and realistically feature
two peaks: one representing Vesuvius' cone, and the other being the remnants of the Somma strato-volcano. These drawings
testify to the interest and good knowledge of the local landscape. In most stories, however, the imaginary volcano is located
in remote regions, surrounded perhaps by a few isolated houses and, in general, at safe distance from populated areas. Only in
a few cases the volcano is portrayed in an urban environment.
Volcanic unrest is marked by a number of realistic precursors, such as shallow seismicity or the emission of smelly volcanic
gases, and is often associated with noise. Interestingly, in some cases, the first signs of volcanic unrest are actual eruptive





events, involving explosions and the launch of ejecta. In general, the unrest phase is too short to take action before the volcano
erupts. The quick transition from unrest to eruption suggests that most stories feature volcanoes with an open conduit. Open-
conduit volcanoes erupt more frequently and their activity is more likely to appear on television or social media. These images
easily contribute to building our mental model of erupting volcanoes. The eruptions from open conduit volcanoes are easily
strombolian, featuring launches of volcanic bombs, lava flows and spectacular lava fountains that closely resemble the events
drawn by the kids.
Most imaginary eruptions are small events, if seen through the eyes of a volcanologist. The main feature is usually a lava flow
that propagates along the slope of the volcanic cone. This effusion is commonly accompanied by the emission of gas and by
the launch of lapilli and bombs that in a few cases may reach beyond the volcano's slopes. This eruptive style recalls common
footage from frequently active Etna or Stromboli volcanoes, in Italy. Some stories mention ash, and this may reflect family
anecdotal accounts of last Vesuvius' eruption, in 1944. A small number of drawings show the development of an eruptive
column (that is never mentioned in the written text). The height of the column is usually small compared to the size of the
volcanic edifice. Only in a few cases, the drawing suggests that the eruption impacted a wider area (i.e., the entire city). Most
stories provide little or no evidence to assess the duration of the eruption. When they do, the event is short-lasting and usually
ends within a few hours or a day.
The imaginary eruption has consequences: most stories describe burnt vegetation and damages to houses and roads. In some
stories people are hurt or killed. Almost half of the stories (42%) mention casualties, but most stories reflect the optimistic bias
already seen in the analysis of risk perception conducted among adults. In these cases, the protagonists survive even though
others are severely affected. Ash covering the landscape and causing respiratory problems is also mentioned a few times.
Damage may be limited (especially when the eruption itself is small), but in a few stories destruction is pervasive. In many
cases, the eruption's consequences are long-lasting, and affect the lifestyle of the characters involved.
Interestingly, in a few stories the characters are killed or injured by events that have nothing to do with the eruption, suggesting
a clear understanding of the multiple hazards that threatens our communities.
The stories collected in Stromboli reveal a strong relation with the volcano and a good knowledge of its various eruptive styles
and products. The two stories that feature the volcano itself as a protagonist both suggest a strong tie connecting Stromboli
with its islet Strombolicchio and with the other Aeolian islands. This is consistent with the geological evolution of the
archipelago. In general, both the drawings and the written descriptions of the stories reveal a close attention toward eruptive
phenomena and their consequences.
As mentioned in section 2, some of the stories (39) were collected in schools located in non volcanic areas. We did not perform
a systematic comparison of stories drawn in different regions, but we can say that those collected in non volcanic areas often
lack details in the drawings and descriptions of volcanic activity, both before and during the eruption. The presence of
suspended ash, which can cause coughing and hinder respiration, is only mentioned in stories collected in the Neapolitan area,
and may reflect familial accounts of the 1944 Vesuvius eruption.

### 5.2 The people

The characters who live or find themselves near the volcano are commonly alone, or with a single companion. In most cases,
they face the unrest and the eruption without the support of a wide community. The protagonists are mostly well aware of the
impending danger, and discuss their options and fears with friends or neighbours. The stories provide a very realistic picture
of people's behaviours, highlighting well known issues, such as warning signs or alerts that are met with disbelief and lack of
action. The stories also capture both the fascination and the fear for the natural phenomenon, as major drivers for people's
actions. While most run or seek shelter, some are paralyzed by fear and a few reckless are rather attracted than scared by the
volcanic activity. The struggle to decide whether or not to leave is also present. The need to move somewhere else to be safe





is a recurrent concept, likely reflecting some knowledge of the emergency plans for the Neapolitan area. However, leaving is
always depicted as a personal decision that not everybody is willing to make.
Moving away from the volcano as a safety measure is described with different nuances in different geographic areas: kids
living far away from actual volcanoes may describe the departure with relief, as a permanent solution to the problem with no
apparent drawbacks; kids from the Neapolitan area, on the other hand, seem to be sorely aware of the many difficulties
associated with leaving, and often describe the characters as sad, lonely and homesick.
A good perception of the complexity of life on an active volcanic system also emerges in a couple of stories from Stromboli,
where people's concerns are mentioned as well as the necessity to leave to be safer elsewhere. One story addresses the very
different perceptions that different people may have of the same phenomenon, ranging from admiration to fear.
In general, the departure is not a planned evacuation, organized and carried out before the eruption, but rather an escape from
the ongoing phenomenon. In several cases, it takes place after the eruption ended, not as a defensive measure but because
houses are damaged, as it happens in case of earthquakes. In the (rare) description of rescue teams, they also intervene in the
aftermath of the eruption. In a seismic country like Italy, images of rescue teams at work after major seismic events is
unfortunately a rather common sight that easily entered the mental model of the kids.
An organized approach to hazard assessment and mitigation is missing. There are no monitoring networks to capture signals,
nor experts capable of interpreting them. If scientists are at the scene, they mostly acknowledge the ongoing activity, without
providing further information, or useful advice. Public authorities are rarely mentioned (less than 5% of the stories), and there
is no coordinated, public response to the change in the volcano's state of activity. Most of the characters face the impending
danger on their own and if action is taken to lower the risk, it mostly happens thanks to individual initiative.
In a small fraction of stories (approximately 10%), the aftermath of the eruption is characterized by reconstruction that sees a
full involvement of the entire community.
**6 Conclusions and steps forward**
The kids and youngsters attending the Imaginary Eruption workshop revealed sharp eyes and a keen attention to the dynamics
of both the natural phenomena and human interactions. Their works, considered together as a whole, capture most of the key
issues related to hazard assessment and mitigation. These were discussed at length in the meetings with the teachers after
working with the students.
The outcome of an eruption does not only depend on the magnitude and explosivity of the volcanic event: people play an
important role. Among other features, consequences depend on the distance between the volcano and inhabited areas, and on
the time available to evacuate. Widespread awareness and preparedness among the population can make a huge difference in
promoting safe actions and mitigating the damages. The conclusion of the story always depends on what goes on in the
preceding frames. The kids' drawings well represent the wide spectrum of possible combinations of eruptive styles and sizes
and people's behaviours.
The collected stories are works of fiction, and do not necessarily represent the actual beliefs or mental model of the drawers.
Nevertheless, in setting up the scene, the students made use of their personal knowledge, and the stories reveal what they think
could get their protagonists in trouble. The analysis of individual stories can be used to identify sound elements of their
understanding of volcanic eruptions and point at topics that may deserve further attention in future outreach work.

Among the positive elements, kids are aware that there will be precursors to an eruption in the Neapolitan area, and can name
several of them. On the other hand, the stories typically describe a very short unrest phase, with macroscopic signals that are
detected shortly before the actual onset of the eruption. The stories do not capture the uncertainty associated with long-lasting
unrest periods, featuring signals whose interpretation may be difficult or controversial.  A long-lasting unrest has been going



on at Campi Flegrei since 2012, but uncertainty on the short-term evolution could characterize the awakening of any dormant
volcano. This crucial phase requires a continuous, strenuous effort to find an acceptable balance between costs and benefits of
possible mitigation actions. The unrest phase causes great stress in the resident population and extreme difficulties in managing
volcanic crises. Yet, the description of the unrest phase is easily neglected in the customary concise descriptions that classifies
volcanoes as dormant or active. School books depict the two options with clear details and images, but perhaps the concept
that a dormant volcano can indeed become active again should be emphasized some more. New outreach materials could focus
on what it takes, and how long it takes, to actually reactivate a dormant volcano. Dealing with uncertainty is hard on emergency
managers and citizens. Being aware that a period of uncertainty is ahead of us is a first step to get ready for it, and possibly
take actions to mitigate the fatigue associated with it. Some volcanic unrests terminate without culminating in an actual
eruption. This has occurred at Campi Flegrei several times (Del Gaudio et al., 2010) and has important implications for hazard
assessment: if an eruption is not the only possible outcome of  precursory signals, false alarms based on monitoring signals
are bound to happen. This possibility is never mentioned in the collected stories, and should be perhaps better emphasized
when discussing volcanic hazards.
When the imaginary eruption strikes, it is usually small. The drawings may in part reflect images and sketches that students
find in their books, where the various features of explosive eruptions are concentrated in a small space for publishing
constraints. We may collaborate with graphic designers to devise sketches of volcanic eruptions that provide a better
understanding of the actual size of big explosive events. And we can put more emphasis on the fact that the same volcano can
display very different eruptive styles and generate big and small events.
The frame describing the eruption is also the one when the protagonists of most stories decide to take action. The kids expect
that an eruption will have a great impact and know that it will be necessary to run or seek shelter. Outreach material and
education itineraries should emphasize that there is time for evacuation, and this time is before the actual eruption begins.
Although distressing, a prolonged unrest phase is just what grants us enough time to organize an orderly evacuation.
As mentioned above, the characters of the Imaginary Eruption do not evacuate, but try to escape, while others hide inside huts
or other improbable shelters. The decision is never planned ahead, but is made in the heat of the moment. In making these
decisions, the characters of the stories mostly rely on their families. In reality, plans for an organized evacuation exist for
exposed areas (red and yellow zones), as detailed in the National Civil Protection Plans for Vesuvius and Campi Flegrei.
However, in a social context where family's ties are strong, outreach activities should take into account that important decisions
will be made inside the families. Education efforts should target all family members, helping the kids to identify unsafe
behaviour and the adults to recognize the optimistic bias.
A small but significant number of stories describe efforts to stop the eruption from happening. While we certainly cannot
control volcanic activity, there are common actions that are taken to hinder lava flow propagation, either using cold water or
building dams to temporarily contain or divert the lava. The problem-solving attitude should be encouraged by showing the
kids that their thinking was very much consistent with existing mitigation strategies. At the same time, we must emphasize
that in case of large explosive eruptions evacuation is the only viable option.
In a few cases, the characters refer to scientists for help, but the nature of the support provided is somewhat blurred. We can
work to better emphasize the insights we gain from volcano monitoring and from experience gathered at different volcanoes.
This knowledge provides the information based on which mitigation actions such as evacuation can be taken. Volcanic gases
are recognized as an integral part of the volcanic activity, but gas is commonly confused with smoke. Specific outreach material
should stress the differences between volcanic gas and smoke, and provide information on the key role of gases in hazard
assessment.
Finally, a few stories reveal the fear that scientists or authorities will not listen, while most indicate that in case of extreme
danger our characters only rely on family and friends. This stresses the need to reinforce the bond between scientists,
emergency managers and the population exposed to volcanic hazards. All the initiatives such as science fairs, citizen science





programs, evacuation exercises, or community meetings bring these stakeholders together and contribute to build and reinforce
mutual trust and understanding.
The Imaginary Eruption has been an interesting exercise that provided us with an unusual insight into how volcanic eruptions
are perceived by kids and youngsters. Far from being a formal assessment of the students' understanding of volcanology, these
stories provided interesting cues to discuss volcanic activities with the kids and explore a wide range of emotions and
sentiments stirred by the thought of an impending eruption. The teachers involved mostly found this a valuable tool to approach
a complex topic, and to build the lecture around the themes that the students propose in their stories. On our side, we gained
very important hints on how to improve our outreach materials and the activities we propose to the schools.
This approach can be easily implemented to explore the mental model related to different natural hazards. We proposed to the
schools of various Italian regions "Imaginary Earthquake" workshops to address seismic risk. During the Edurisk activities
(Camassi et al., 2021) we learned to constantly adapt our approaches and procedures during the projects in the schools, and
when the Covid-induced restrictions stimulated to adapt our teaching activities to the new constraints, we experienced closer
and immediate contact with students and teachers. The tools we used in attendance have been adapted to remote teaching: the
activity was structured in the same way, throughout a 4 sheets story, drawn and written, and we managed to keep the same
style of active participation, built on stimuli and discussions. The results we have obtained, in terms of students' laboratory
restitutions, are fully comparable with those above described obtained from the students of the Neapolitan area. We believe
that this approach could be further extended to other natural hazards.

**Author contribution**

All authors contributed to devising the workshop structure. RC provided the funding to sustain the activities and together with
EE and VP granted the synergy with the EDURISK project. Workshops were held by FB, DM, EE, RC, RN, VP. Analysis of
the collected stories was carried out by MT, FB, EE, DM and results were discussed by all co-authors. MT prepared the
manuscript with contributions from all co-authors.

**Competing interests**

The authors declare that they have no conflict of interest.

**Acknowledgements**

This work was carried out within the EDURISK project that benefited from funding provided by the Italian Presidenza del
Consiglio dei Ministri - Dipartimento della Protezione Civile (DPC). Results and conclusions reported in the manuscript do
not necessarily represent DPC official opinion and policies. We wish to thank all the teachers and students involved in the
workshops for their enthusiastic involvement and response.

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
