# Peer review of "The imaginary eruption - Volcanic activity through kids' eyes 1"

_Geoscience Communication, 2022_

## Referee Comment (RC1)

**Reviewer Comments for 'The Imaginary eruption. Volcanic Activity Through Kids' Eyes**

**General Comments:**

The article offers a unique perspective about student knowledge of volcano hazards, and plenty of results to ponder. It gave this reviewer some new ideas about how the profession might advance its collaboration with curriculum developers, and how to design content for teacher workshops.   The article addresses relevant scientific questions within the scope of Geoscience Communications.  It reports on a novel approach for exploring student knowledge of volcanic eruptions.  The scientific methods and assumptions for the greatest part are clearly outlined.  Its results are plenty sufficient to support the outcomes and conclusions.  This reviewer recommends additional citations in the Introduction, and the citations of other studies of student knowledge.   The title and abstract reflect well the subject of the paper.   The overall presentation of the work is clear and well structured; its language is precise.  This reviewer has made minor recommendations for wording choices to allow full understanding and content appreciation by English-speaking readers.

**Specific Comments:**

Title: Consider a dash rather than period within it.

Sentences ending on lines 29, 30, 31, 32, 36, and 39 require the support of citations. Some recommendations that support text about the complexity of preparedness:

> Chester DK (2005) Volcanoes, society, and culture. In: Marti J, Ernst GJ (eds) Volcanoes and the environment. Cambridge University Press, New York, pp 404–439

> Gregg CE, Houghton BF, Johnston DM, Paton D, Swanson DA (2004) The perception of volcanic risk in Kona communities from Mauna Loa and Hualalai volcanoes, Hawai'i. J Volcanol Geotherm Res 130:179–196 https://doi. org/10.1016/S0377-0273(03)00288-9

Line 79 Please provide a sentence or two about curricula used by the teachers in the longterm, *prior to* the introduction of this activity?  Do all teachers use the same curricula? A general statement about this will be helpful to the reader.

Somewhere in the Introduction, consider an additional one or two sentences about the importance of school children in bringing hazard and preparedness messages home to the parents.  The children are the conduits of knowledge to the extended family.  Additionally, it seems that some connection could be made between the 2008 risk perception study and this new study (just an idea).

> Barberi F, Davis MS, Isaia R, Nave R, Ricci T (2008) Volcanic risk perception in the Vesuvius population. J Volcanol Geotherm Res 172:244–258. https://doi.org/ 10.1016/j.jvolgeores.2007.12.011

> Cardona O (1997) Management of the volcanic crises of Galeras volcano: social, economic and institutional aspects. J Volcanol Geotherm Res 7:313–324. https://doi.org/10.1016/S0377-0273(96)00102-3

> Carlino S, Somma R, Mayberry G (2008) Volcanic risk perception of young people in the urban areas of Vesuvius—comparisons with other volcanic areas and implications. Volcanol Geotherm Res 172(3,4):229–243. https://doi.org/10. 1016/j.jvolgeores.2007.12.010

Johnston D, Becker, Jl Coomer, M Ronan, K Davis, M Gregg, C (2006) Children's risk perceptions and preparedness: Mt Rainier 2006 hazard education assessment tabulated results. GNS Science Report 2006, 16 June: 30 https:// scholar.dominican.edu/all-faculty/177/

Ronan K, Johnston D (2005) Promoting community resilience in disasters: The role for schools, youth, and families. Springer, New York NY

**Technical Corrections:**
Line 36-39  This is a run-on sentence.  Split the sentence, and make the meaning clear.
Begin the sentence with The most recent (certainly not the last).
Separate Vesuvius and Ischia into two sentences to reduce confusion.   It is also a run-on sentence.
Citation needed.
Sentence edit recommended: 'volcanic eruptions, by forming a collaboration with the local schools…'
Suggested: 'On Stromboli, we gathered 11 stories, 9 of which were completed…'
For clarity in English, the word realization could be replaced with *Completion*, or *Unfolding*, or *Advancement*.
*lesson* (singular)
The analysis provides clues *about* volcanic risks and what *factors* amplify or mitigate them.
For clarity:   'During the analysis we used all of the considerations raised…'
Change Indications to *Instructions.*
Delete *also*.  Realization to *Completion* of each frame.
Realize to *complete;* indications to *instructions.*
(or when the allotted time passed), *did* we *provide* information
recommended:  We collected 190 stories with text (2 stories without written text), and with drawings…
recommended: change corpus to *assemblage*
death or salvation:  salvation does not translate well.  Does it mean injured and alive?  People complete *unaffected*?
Just a note that it is interesting how the students are often the protagonists in their own stories!
change buttered to *battered.*
remove *really*
lava *flows*
Sentence structure: Both events *raised* an eruption column…and *generated* pyroclastic flows…
413-414 *non-volcanic*
473-475  This concept of text book error or generality is very important.  It is good to see it emphasized here.  Later in Conclusions, you could note that scientists and textbook writers should become acquainted and agree to use terms that are not so simplistic.
These are other important points.  Consider adding that graphics could demonstrate the *common sequence of events*—both precursory events, and during the eruption.
The concept of the family preparing together is of greatest importance.  Authors could add some specific recommendations, such as the family having a scavenger hunt to find items for their emergency kit, working together to complete family communication plans, and similar.

Bravo!  This is a very nicely executed study, and a well-structured report.

---

## Author Response (AR1)

**Response to the first review**

We thank the reviewer for the appreciation of our work and for the useful suggestions provided that improve the original manuscript.
We followed all the recommendations and modified the manuscript accordingly. In particular, we added all the references suggested by the reviewer and introduced some more where requested.
We provided some information on Italian schools curricula and emphasized the role of kids in bringing crucial information to their families.
We also introduced all the technical corrections proposed by the reviewer.

**Response to Sam Illingworth's review**

We thank the reviewer for his kind opinion about our work and for the useful suggestions provided. Below we address in italic each of the points he made, and highlight where the original text has been changed.

- The abstract is on the whole excellent. However, please consider the overuse of the word 'interesting' which appears quite a lot over the course of only a few sentences.

Touché. The word interesting is just so easy to use. We just got rid of it. Most of the times.

- The biggest issue that needs to be addressed with this study is one of ethics. What ethical clearance did this study receive? How did the participants give their informed consent to participate in this study (this is especially important as they are a potentially vulnerable audience)? What safeguarding and other issues arose? And how were these mitigated by the research team. See Section 4.1 of Archer (2021) or Section 2.2 of Mohadjer (2021) for examples of how you might best include this in your manuscript.

This is a good point. As mentioned in the manuscript, The Imaginary Eruption workshop was carried out within the framework of the Edurisk Project, proposed to all Italian schools. The participation of each school into the project follows an official procedure that also ensures ethical clearance. We added a few lines of text to better emphasize this important aspect.

- It was not entirely clear how you moved from the frequency of words (Section 4.1) to the framing of these occurrences (Section 4.2). Could you please include more detail about how this was done and what method was adopted, as at the moment it would be difficult for an independent researcher to repeat your findings, or even for a new researcher to adopt this work for their own study.

The analysis carried out in Section 4.1 only accounts for the written part of the stories, and considers all text, written by all students in all frames, all together as a single dataset from which frequencies are considered. In the following chapter 4.2, the analysis considers the collection of stories that are analyzed and compared, frame by frame. We added a few sentences to clarify that.

- I wonder if Section 4.3 on Stromboli could be removed entirely, as it does not add anything significantly to the study, and as you point out yourselves is much less developed than the work in the other 10 school districts.

We cannot deny that Section 4.3 does not present results as detailed as those obtained in the Neapolitan area. Nevertheless, these few kids did witness a real explosive eruption. We consider this a valuable addition to the overall picture and would prefer to maintain the Section.

- The Conclusions are overly long and could could benefit from both a streamlining and also a reframing around recommendations and/or advise for others wanting to adapt /build on this approach

We modified the Conclusions to make them easier to read. We hope we succeeded..

- For the most part this is a very well-written proposal, but there are several technical corrections that need to be addressed. I picked up all the same ones as Reviewer 1, and so will not repeat them here. A final close proofread and edit after all these changes have been made would also be appreciated.

We introduced all the suggestions provided by Reviewer #1, and carefully checked the revised manuscript. Hopefully, we could capture most of the errors.

---

## Author Response (AR2)

We thank the Editor for the attention to our work.
We addressed all the comments and modified the manuscript and Supplements' captions according to the suggestions.